# MoDA: Modulation Adapter for Fine-Grained Visual Understanding in Instructional MLLMs

Wayner Barrios [1]  Andrés Villa [2]  Juan C. Leon Alcazar [2]  SouYoung Jin [1 †]  Bernard Ghanem [2 †]

## Abstract

Multimodal Large Language Models (MLLMs) have achieved remarkable success in instruction-following tasks by integrating pretrained visual encoders with large language models (LLMs). However, existing approaches often struggle with fine-grained visual grounding due to semantic entanglement in visual patch representations, where individual patches blend multiple distinct visual elements, making it difficult for models to focus on instruction-relevant details. To address this challenge, we propose MoDA (Modulation Adapter), a lightweight module that enhances visual grounding through instruction-guided channel-wise modulation. Unlike token-level methods such as Q-Former that perform additive feature selection, MoDA operates at the channel level through multiplicative modulation on already-aligned features, enabling fine-grained control over which embedding dimensions are relevant for each instruction. Following the standard LLaVA training protocol, MoDA applies cross-attention between language instructions and pre-aligned visual features, generating dynamic modulation masks without architectural modifications or additional supervision. We evaluate MoDA across 12 benchmarks spanning visual question answering, vision-centric reasoning, and hallucination detection, including recent 2024 benchmarks (MMVP, CV-Bench, MMStar, RealWorldQA), on three distinct MLLM architectures: LLaVA-1.5, LLaVA-MoRE (2025), and Qwen3-VL (2025). MoDA delivers consistent gains across all three families, with +12.0 points on MMVP for the LLaVA-1.5 family and +4.8 points

on ScienceQA for the LLaVA-MoRE family, and +4.9 ScienceQA, +4.1 RealWorldQA, and +3.8 GQA on Qwen3-VL, confirming that the gains generalize beyond CLIP-based encoders with minimal overhead ($< 1\%$ FLOPs). Code is available at https://github.com/waybarrios/MoDA.

## 1. Introduction

The rapid progress of Large Language Models (LLMs) has led to impressive zero-shot performance across a broad spectrum of natural language processing benchmarks (Wang et al., 2024; Chung et al., 2024; Liang et al., 2023; Llama Team, AI @ Meta, 2024; Yang et al., 2024; Team, 2025). The success of instruction-tuned LLMs has driven computer vision research in a similar direction, ultimately leading to the development of Multimodal Large Language Models (MLLMs). MLLMs integrate pretrained visual encoders with large language models via lightweight adapter modules, enabling efficient cross-modal alignment and strong performance across diverse multimodal tasks, including Visual Question Answering (VQA), Image Captioning, Image Reasoning, and Image Classification.

Despite their success, state-of-the-art MLLMs frequently struggle with fine-grained visual understanding, particularly when answering queries that require precise localization and detailed reasoning about specific visual elements. This limitation manifests as hallucinations, where model outputs contradict actual image semantics, undermining reliability in real-world applications. Prior analyses have identified the CLIP-based visual encoder as a key bottleneck: its patch-based representations often fail to capture localized details due to semantic entanglement within individual patches (Villa et al., 2024; Tong et al., 2024b; Kar et al., 2024). While some works incorporate multiple specialized visual encoders (Tong et al., 2024b; Kar et al., 2024) or fine-tune CLIP for better local structure preservation (Villa et al., 2025), these approaches often introduce substantial computational overhead or require large-scale retraining.

We illustrate this semantic entanglement problem through a practical example. Figure 1a shows a $3 \times 3$ grid over a sleeping French bulldog with a plush toy, simulating CLIP's

[†]Equal senior authorship. [1]Department of Computer Science, Dartmouth College, Hanover, NH, USA [2]King Abdullah University of Science and Technology (KAUST), Thuwal, Saudi Arabia. Correspondence to: Wayner Barrios <wayner.j.barrios.quiroga.gr@dartmouth.edu>.

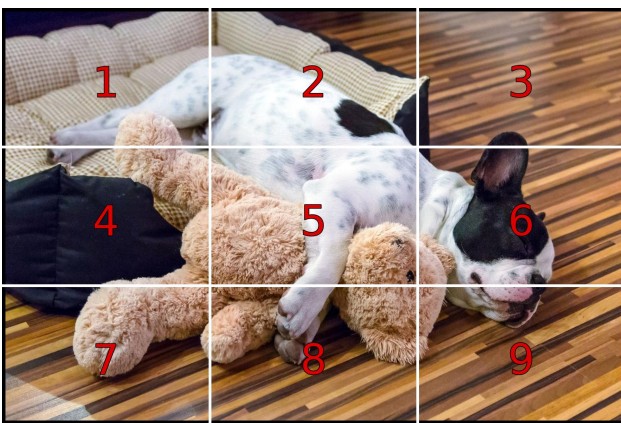
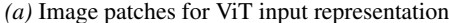

*(a)* Image patches for ViT input representation

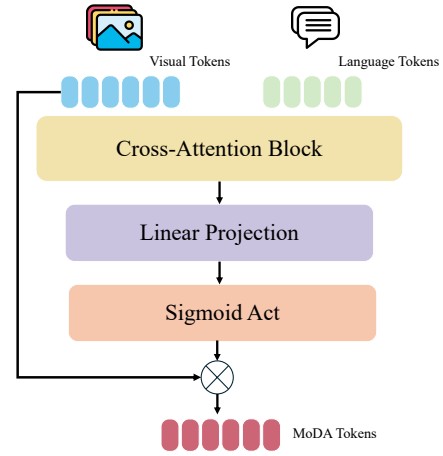

*(b)* Modulation Adapter (MoDA) Architecture

*Figure 1.* **ViT patch representation and our proposed Modulation Adapter (MoDA). (a)** ViT splits images into fixed-size patches, each projected into high-dimensional embeddings. This partitioning blends semantically distinct elements (e.g., dog, toy, floor within a single patch), creating entangled representations. **(b)** MoDA is a lightweight module that modulates visual embeddings via cross-attention using language tokens as guidance, enabling selective attention without architectural modifications or additional supervision.

visual tokenization with enlarged patches for visualization. Crucially, none of the patches contain uniform visual elements. Patch 5 blends the dog's torso, stuffed toy, and cushioned bed; patch 6 mixes the dog's head, ear, and hardwood floor. This forces the visual encoder to combine distinct shapes, textures, and semantic concepts into single embeddings, where individual feature dimensions encode multiple unrelated meanings (Oquab et al., 2024; Ma et al., 2022; Zhou et al., 2024; Shi et al., 2024). Consequently, when processing language queries like *"What color is the dog's ear?"* or *"Is the toy lying on the bed or the floor?"*, the model must disentangle mixed visual representations to provide reliable answers, often failing to focus on instruction-relevant details.

Existing approaches to address this challenge fall into several categories. Some works apply attention masking techniques adapted from NLP (Fan et al., 2021; Tang et al., 2021; Lin & Joe, 2023), but these typically operate on token-level sparsity rather than channel-wise feature refinement. Others employ layer-wise adaptive masking (Barrios & Jin, 2024), which introduces substantial overhead when applied to deep models. Most critically, these approaches lack instruction-guided conditioning, missing the opportunity to dynamically adapt visual attention based on specific language queries. This leads to our central question: *How can we enable MLLMs to dynamically focus on instruction-relevant visual details for better visual understanding without architectural modifications or computational overhead?*

We address this challenge through the *Modulation Adapter* (**MoDA**), a lightweight module that performs instruction-guided channel-wise modulation of pre-aligned visual features. MoDA differs fundamentally from existing cross-

attention methods like Q-Former (Li et al., 2023a) and InstructBLIP (Dai et al., 2023) in three key aspects: (1) *granularity*: MoDA operates at the channel level within each token's embedding dimensions, rather than selecting or reweighting entire tokens; (2) *operation*: MoDA applies multiplicative modulation through element-wise Hadamard products, enabling selective suppression of irrelevant dimensions, unlike additive residual approaches; and (3) *position*: MoDA operates post-alignment on already-projected features, complementing rather than replacing existing adapters. Our approach employs cross-attention between language instructions and visual features to generate dynamic modulation masks, enabling precise visual-language alignment without modifying the underlying MLLM architecture. Crucially, MoDA's effectiveness scales with visual encoder quality: while providing modest improvements with standard CLIP encoders, it achieves substantial gains when paired with richer representations like SigLIP-S2, demonstrating that instruction-guided modulation becomes increasingly valuable for fine-grained visual understanding. MoDA integrates seamlessly into existing two-stage instruction-tuning pipelines, requires no additional supervision or training data, and introduces minimal computational overhead ($< 1\%$ FLOPs, $3.7\%$ parameters).

We validate MoDA across 12 diverse benchmarks spanning visual question answering, vision-centric reasoning, and hallucination detection. We integrate MoDA into three distinct MLLM architectures: LLaVA-1.5 (Liu et al., 2024), LLaVA-MoRE (Cocchi et al., 2025) (2025), and Qwen3-VL (Qwen Team, 2025) (2025), the latter a recent state-of-the-art MLLM whose native ViT departs from the CLIP family. We evaluate on recent benchmarks including MMVP,

CV-Bench, MMStar, and RealWorldQA. MoDA delivers **consistent improvements across all three families**, with **+12.0 points** on MMVP for the LLaVA-1.5 family and **+4.8 points** on ScienceQA for the LLaVA-MoRE family, and **+4.9 ScienceQA**, **+4.1 RealWorldQA**, and **+3.8 GQA** on Qwen3-VL, confirming that the gains generalize beyond CLIP-based encoders. Ablation studies confirm these gains stem from architectural design rather than parameter scaling, with strongest improvements on fine-grained visual tasks. Our main contributions are: **(i)** identifying semantic entanglement in visual patch representations and proposing MoDA, a novel instruction-guided channel-wise modulation approach that addresses this limitation; **(ii)** demonstrating substantial performance improvements with minimal computational overhead, adding only $< 1\%$ FLOPs while achieving consistent gains across diverse benchmarks; and **(iii)** comprehensive evaluation showing MoDA's effectiveness stems from architectural innovation rather than capacity increases.

## 2. Related Work

**Multimodal Instruction Tuning.** Instruction-tuning has become the standard approach for enhancing MLLMs by incorporating task-specific natural language commands that improve generalization across vision-language tasks. The typical pipeline involves two stages: first, cross-modal alignment projects visual features from encoders like CLIP (Liu et al., 2023a; 2024; Cocchi et al., 2025; Chen et al., 2024a) or Q-Former (Li et al., 2023a; Dai et al., 2023) into the language embedding space; second, instruction-following fine-tuning enhances task generalization. Our approach builds upon the second stage, assuming well-aligned multimodal representations and focusing on instruction-conditioned refinement of visual features.

**Cross-Modal Attention and Feature Aggregation.** Modern MLLMs increasingly leverage cross-attention mechanisms for multimodal integration. InstructBLIP (Dai et al., 2023) pioneered injecting language queries directly into Q-Former architecture for selective visual attention, while Cambrian-1 (Tong et al., 2024a) employs cross-attention at the token level for multimodal reasoning. Other approaches explore multiple visual encoders with cross-attention fusion (Kar et al., 2024) or learnable query tokens for task-relevant information extraction. However, these methods primarily operate on discrete token interactions. MoDA differs by introducing channel-wise modulation through cross-attention, where language instructions guide the re-weighting of continuous feature dimensions rather than discrete tokens, enabling fine-grained semantic control while preserving the spatial structure of visual representations.

**Attention Masking and Multimodal Efficiency.** Attention masking strategies in multimodal models can be categorized into three main paradigms. Token-level sparsity

methods like SwinBERT (Lin et al., 2022) generate fixed sparse masks at input, trading adaptability for efficiency. Layer-wise adaptive approaches such as LAM (Barrios & Jin, 2024) recompute learnable masks at each transformer layer, enabling dynamic attention but introducing computational overhead that scales problematically with network depth. Visual-only mechanisms like MST (Li et al., 2021) perform attention-guided masking within the vision encoder without language interaction. MoDA introduces a distinct rationale through single-pass channel-wise modulation that operates on continuous feature dimensions rather than discrete tokens, performs modulation only once after the adapter stage to avoid scaling issues, and explicitly incorporates language guidance for instruction-conditioned refinement.

**Adapter Architectures.** Adapter modules serve as crucial interfaces between visual encoders and language models in MLLMs. While LLaVA-family models (Liu et al., 2023a; Cocchi et al., 2025; Chen et al., 2024a) employ lightweight adapters for efficient CLIP-to-language mapping, recent innovations include attention pooling and multi-scale feature aggregation. However, these approaches primarily focus on initial cross-modal alignment rather than dynamic, instruction-conditioned refinement. MoDA complements existing adapter architectures by operating as a post-processing module that refines already-aligned features based on specific language instructions, maintaining compatibility with standard MLLM designs while providing targeted improvements in fine-grained visual grounding.

**Visual Feature Refinement Across the Pipeline.** Recent work has explored visual feature refinement at different stages of the MLLM pipeline. At the encoder level, EAGLE (Villa et al., 2025) fine-tunes CLIP to better preserve local structure, requiring additional pre-training. Instruction-Guided Fusion (Li, 2025) addresses layer selection by dynamically weighting features from different encoder depths based on task requirements. At the decoder level, MoReS (Bi et al., 2024) applies linear transformations at each LLM layer to address modality imbalance where text dominates visual representations. AdaLink (Wang et al., 2023) introduces input-centric parameter-efficient fine-tuning through non-intrusive adaptation mechanisms. These methods operate at distinct pipeline stages: encoder pre-training, layer selection, or per-layer LLM transformations. In contrast, MoDA operates at the adapter-to-LLM interface, performing channel-wise modulation on already-aligned features before they enter the language model. This positioning makes MoDA potentially complementary to the above approaches, as improved encoder features or layer selection could provide higher-quality inputs for MoDA's channel-wise refinement, while MoDA's instruction-conditioned modulation could enhance the features before downstream processing by methods operating within the LLM.

## 3. Visual Feature Modulation

MoDA (MODulation Adapter) is a lightweight module designed to post-process visual embeddings from an MLLM's adapter. MoDA leverages the alignment of visual and language embedding spaces, and selects the most relevant visual features based on the input language query. Our module assigns individual weights to these visual features through cross-attention with the language embedding, these weights are encoded in a soft modulation mask. This mask promotes relevant visual embedding dimensions while de-emphasizing less relevant ones. The resulting re-weighted features are then passed to the LLM for decoding.

Within a MLLM, the MoDA component is integrated after the pre-trained adapter. Given a pre-aligned visual feature map $V_{\text{aligned}}$, our objective is to learn a function $F(\cdot)$ that estimates a modulation operator based on the current text query $T$. This operator is then applied element-wise across the embedding dimensions of the visual features, as follows:

$$\widetilde{V}_{\text{aligned}} = V_{\text{aligned}} \odot F(T, V_{\text{aligned}}) \tag{1}$$

Where $\odot$ denotes the Hadamard product along the embedding dimension. The function $F(T, V_{\text{aligned}})$ is dependent on the text prompt, therefore, it modulates the attention of the MLLM towards the more informative embeddings according to the current text prompt. As a consequence, the re-weighted feature map $\widetilde{V}_{\text{aligned}}$ provides refined visual cues, which improve the MLLM's ability to resolve the complex natural language instructions in modern MLLM benchmarks.

### 3.1. Modulation Adapter (MoDA) Design

Let $V_{\text{aligned}} \in \mathbb{R}^{B \times N \times E}$ denote the language aligned visual features obtained from the adapter module of the MLLM, where $B$ is the batch size, $N$ is the number of image tokens, and $E$ is the embedding dimension. Let $T \in \mathbb{R}^{B \times M \times E}$ represent the language token embeddings, where $M$ is the number of text tokens. The $T$ embeddings are obtained directly from the initial layers of the LLM component. MoDA learns a modulation function $F(\cdot) \in [0,1]^E$ conditioned on the multi-modal feature embedding $\{V_{\text{aligned}}, T\}$, followed by a linear projection and sigmoid activation. The re-weighted visual features $\widetilde{V}_{\text{aligned}}$ are computed as:

$$\widetilde{V}_{\text{aligned}} = V_{\text{aligned}} \odot \sigma\left(W \cdot F(T, V_{\text{aligned}})\right) \tag{2}$$

The modulation function $F(\cdot)$ is implemented using a stack of Transformer Layers that takes the language-aligned visual features $V_{\text{aligned}}$ as the target sequence and the language token embeddings $T$ as the memory input. The matrix $W \in \mathbb{R}^{E \times E}$ is a learnable linear projection, and $\sigma(\cdot)$ is the sigmoid activation function applied element-wise to constrain the mask values in the range $[0,1]$. In practice, the output of $\sigma(W \cdot F(T, V_{\text{aligned}}))$ can be interpreted as a channel-wise mask that independently re-weights each feature channel in the visual embedding.

The MoDA module consists of multiple cross-attention Transformer layers, each composed of three main components: (i) a multi-head cross-attention mechanism that allows each visual token to attend to relevant parts of the language input, (ii) a feed-forward network that refines the representation at each layer, and (iii) residual connections and layer normalization to facilitate training stability and convergence. After passing through this stack, the output is projected and passed through the sigmoid non-linearity to generate the final modulation mask $\mathcal{M}$. This mask is applied following equation 1 to obtain the refined visual representation $\widetilde{V}_{\text{aligned}}$.

### 3.2. MoDA MLLM Architecture and Training Details

MLLMs incorporating MoDA adopt the architecture and two-stage training protocol introduced in LLaVA (Liu et al., 2023a), which combines a vision encoder with a large language model (LLM). As illustrated in Figure 2, our enhanced MLLM retains the three fundamental components of (Liu et al., 2023a): a vision encoder, an adapter module for visual-language alignment, and a pretrained LLM. However, MoDA (Modulation Adapter) is introduced as a novel component that operates as an interface between the pretrained vision-language adapter and the LLM. Following this integration, the vision encoder extracts patch-level visual features from the input image, which are then projected into the language embedding space by the standard adapter module. MoDA then takes these aligned visual features, estimates channel-wise modulation weights, and passes the modulated features to the LLM for language decoding.

Following the standard practice in LLaVA models, the enhanced visual embeddings are then used as prefix tokens for the LLM. Then, LLM mixes the modulated visual tokens with the input query tokens, and autoregressively generates a natural language response.

**Training Procedure.** The training of MoDA follows the two-stage approach of (Liu et al., 2023a). In the first stage, only the original visual adapter is trained following the LLaVA protocol (Liu et al., 2023a; 2024). The vision encoder and the LLM remain frozen during this phase, and the training is supervised using an autoregressive language modeling objective. The LLM is prompted with language-aligned image features (via the adapter) and a language instruction, and it learns to predict the target output sequence using standard cross-entropy loss over the predicted tokens.

In the second stage, we introduce the MoDA module to

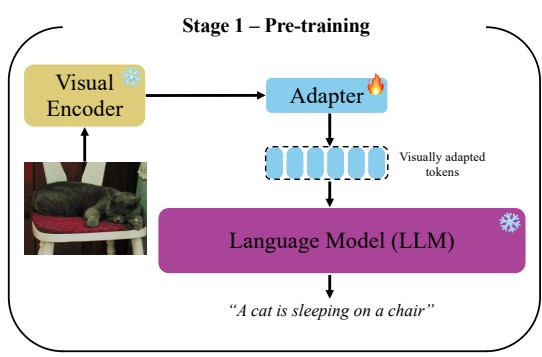

*(a)* Stage 1 - Pre-Training

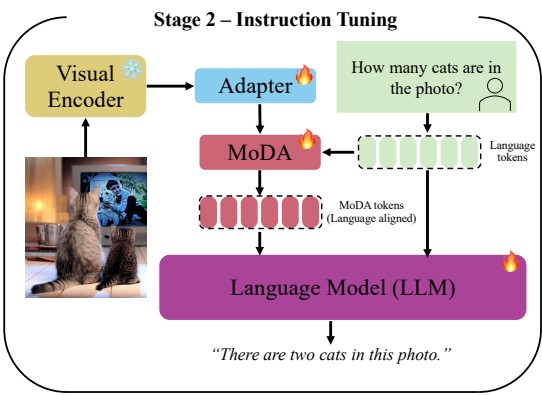

*(b)* Stage 2 - Instruction Tuning

*Figure 2.* **Training Framework.** MoDA follows a two-stage process: **(1) Pre-training** the adapter for visual-language alignment, and **(2) Instruction Tuning** where the adapter and MoDA are fine-tuned with a pretrained LLM. MoDA refines adapter outputs by emphasizing language-relevant visual features.

enhance the model's grounding capabilities. MoDA is initialized using Xavier initialization, while the visual adapter retains the weights learned on the initial stage. During this phase, we finetune both MoDA and the LLM jointly, enabling the model to better attend to semantically relevant visual cues through MoDA while improving its overall conversational ability.

The learning objective across both stages remains the same: given a sequence of input tokens and visual embeddings, the model is trained to minimize the autoregressive cross-entropy loss:

$$\mathcal{L}_{\text{CE}} = -\sum_{t=1}^{|y|} \log P(y_t \mid y_{<t}, \widetilde{V}_{\text{aligned}}, T) \qquad (3)$$

where $y_t$ is the ground-truth token at time step $t$, $y_{<t}$ denotes the previously generated tokens, $\widetilde{V}_{\text{aligned}}$ are the modulated visual features produced by MoDA, and $T$ represents the tokenized instruction.

# 4. Experiments

Our experimental evaluation strategically targets the semantic entanglement problem identified in Figure 1 through 12 benchmarks spanning three categories: hallucination detection where models must distinguish visual evidence from learned priors, complex reasoning requiring precise visual-language coordination, and fine-grained visual analysis demanding detailed instruction-following capabilities.

**Experimental Setup.** We evaluate MoDA across 12 benchmarks spanning visual question answering (GQA, ScienceQA, MMBench, RealWorldQA, ChartQA), vision-

centric tasks (LLaVA-Wild, MM-Vet, MMStar, V*Bench, CV-Bench), and hallucination detection (POPE, MMVP). These benchmarks require strong language capabilities for instruction following and precise visual processing. Our model follows the standard LLaVA architecture with MoDA integrated as a lightweight cross-attention module between the adapter and language model. We adopt the two-stage training protocol of LLaVA-1.5, using the same hyperparameters and training data to ensure fair comparison. Further details are provided in Section A.1.

## 4.1. Results

We evaluate MoDA across 12 benchmarks spanning visual question answering, vision-centric reasoning, and hallucination detection. The overall trend aligns with our motivation (Section 1 and Section 3): by applying cross-attentive channel modulation, MoDA directs information flow toward instruction-relevant features and enables high-capacity encoders to produce more precise and well-grounded outputs.

**VQA Performance.** As shown in Table 1, MoDA improves VQA by transforming the instruction into a soft, channel-wise mask over visual embeddings. Within the LLaVA-MoRE family, the gains scale with encoder quality. With SigLIP-S2, ScienceQA increases by 4.8 points, from 77.1 to 81.9, and MoDA attains the strongest LLaVA-MoRE scores on all five VQA benchmarks: GQA at 65.4, ScienceQA at 81.9, MMBench at 72.4, RealWorldQA at 58.2, and ChartQA at 18.1. These consistent improvements demonstrate that instruction-conditioned channel modulation effectively guides the model toward task-relevant visual features across diverse question types.

*Table 1.* **Performance on Visual Question Answering benchmarks.** We evaluate three MLLM architectures (LLaVA-1.5, LLaVA-MoRE, and Qwen3-VL) on GQA, ScienceQA, MMBench, RealWorldQA, and ChartQA. **Bold underlined** values indicate highest scores per benchmark. **Bold** values show best performance within each baseline comparison. Gray text indicates models trained on different larger data distributions. All metrics are percentages; higher is better.

| Method | LLM | GQA | ScienceQA | MMBench | RealWorldQA | ChartQA |
|---|---|---|---|---|---|---|
| BLIP-2 (Li et al., 2023a) | FLAN-T5 | 41.0 | 61.0 | - | 22.4 | - |
| InstructBLIP (Dai et al., 2023) | Vicuna-7B | 42.9 | 60.5 | 36.0 | 1.0 | 0.2 |
| Qwen-VL-Chat (Bai et al., 2023) | Qwen-7B | 57.5 | 68.2 | 60.6 | - | - |
| LLaVA (Liu et al., 2023a) | Vicuna-7B | - | 38.5 | 34.1 | 11.0 | - |
| LLaVA-1.5 (Liu et al., 2024) | Vicuna-13B | 63.3 | 71.6 | 67.7 | 45.8 | 17.1 |
| ShareGPT-4V (Chen et al., 2024a) | Vicuna-7B | 63.3 | 68.4 | 68.8 | 52.0 | 16.8 |
| LLaVA-1.5 (Liu et al., 2024) | Vicuna-7B | 62.4 | 69.0 | 64.3 | 44.3 | **17.0** |
| LLaVA-1.5 + **MoDA (ours)** | Vicuna-7B | **62.5** | **71.0** | **64.8** | **53.4** | 13.2 |
| LLaVA-MoRE OpenAI CLIP (Cocchi et al., 2025) | LLaMA 3.1-8B | 63.6 | 76.3 | **72.3** | 57.1 | 15.5 |
| LLaVA-MoRE OpenAI CLIP + **MoDA (ours)** | LLaMA 3.1-8B | **64.4** | **77.8** | 72.0 | **58.0** | **15.6** |
| LLaVA-MoRE SigLIP-S2 (Cocchi et al., 2025) | LLaMA 3.1-8B | 64.9 | 77.1 | 71.8 | 57.2 | 17.3 |
| LLaVA-MoRE SigLIP-S2 + **MoDA (ours)** | LLaMA 3.1-8B | **65.4** | **81.9** | **72.4** | **58.2** | **18.1** |
| Qwen3-VL-2B-Instruct (Qwen Team, 2025) | Qwen3-2B | 59.4 | 79.3 | 86.5 | 64.7 | **80.0** |
| Qwen3-VL-2B-Instruct + **MoDA (ours)** | Qwen3-2B | **63.2** | **84.2** | **87.4** | **68.8** | 79.0 |

*Table 2.* **Performance on vision-centric benchmarks requiring fine-grained visual understanding.** We evaluate three MLLM architectures (LLaVA-1.5, LLaVA-MoRE, and Qwen3-VL) on LLaVA-Wild, MM-Vet, MMStar, V*Bench, and CV-Bench. **Bold underlined** values indicate highest scores per benchmark. **Bold** values show best performance within each baseline comparison. Gray text indicates models trained on different data distributions. A dash (-) indicates a result not reported. All metrics are percentages; higher is better.

| Method | LLM | LLaVA-Wild | MM-Vet | MMStar | V*Bench | CV-Bench |
|---|---|---|---|---|---|---|
| BLIP-2 (Li et al., 2023a) | FLAN-T5 | 38.1 | - | 37.6 | - | - |
| InstructBLIP (Dai et al., 2023) | Vicuna-7B | 60.9 | 26.2 | 1.0 | 34.0 | - |
| Qwen-VL-Chat (Bai et al., 2023) | Qwen-7B | - | - | 37.7 | - | - |
| LLaVA (Liu et al., 2023a) | Vicuna-7B | 62.8 | 23.8 | - | 35.5 | - |
| LLaVA-1.5 (Liu et al., 2024) | Vicuna-13B | 72.5 | - | - | - | 60.9 |
| ShareGPT-4V (Chen et al., 2024a) | Vicuna-7B | 72.6 | - | 33.0 | - | 61.8 |
| LLaVA-1.5 (Liu et al., 2024) | Vicuna-7B | 65.4 | 28.1 | 27.6 | 42.9 | **59.0** |
| LLaVA-1.5 + **MoDA (ours)** | Vicuna-7B | **68.0** | **29.9** | **32.9** | **44.5** | 58.2 |
| LLaVA-MoRE OpenAI CLIP (Cocchi et al., 2025) | LLaMA 3.1-8B | 71.2 | 25.2 | 35.7 | 42.8 | 59.9 |
| LLaVA-MoRE OpenAI CLIP + **MoDA (ours)** | LLaMA 3.1-8B | **73.9** | **26.6** | **36.7** | **44.0** | **61.0** |
| LLaVA-MoRE SigLIP-S2 (Cocchi et al., 2025) | LLaMA 3.1-8B | **72.0** | 27.7 | 35.8 | 44.4 | 61.2 |
| LLaVA-MoRE SigLIP-S2 + **MoDA (ours)** | LLaMA 3.1-8B | 67.6 | **28.3** | **38.5** | **44.8** | **62.2** |
| Qwen3-VL-2B-Instruct (Qwen Team, 2025) | Qwen3-2B | - | 51.9 | 53.9 | **77.0** | 80.9 |
| Qwen3-VL-2B-Instruct + **MoDA (ours)** | Qwen3-2B | - | **52.0** | **55.3** | 74.9 | **81.0** |

*Table 3.* **Hallucination detection benchmarks.** We evaluate three MLLM architectures (LLaVA-1.5, LLaVA-MoRE, and Qwen3-VL) on POPE and MMVP. **Bold underlined**: highest per benchmark. **Bold**: best within baseline. *: Gemma 3 (Team, 2025) grader.

| Method | LLM | POPE | MMVP* |
|---|---|---|---|
| BLIP-2 (Li et al., 2023a) | FLAN-T5 | - | - |
| InstructBLIP (Dai et al., 2023) | Vicuna-7B | 85.0 | 16.9 |
| LLaVA (Liu et al., 2023a) | Vicuna-7B | - | 6.6 |
| LLaVA-1.5 (Liu et al., 2024) | Vicuna-13B | 85.9 | 24.7 |
| LLaVA-1.5 (Liu et al., 2024) | Vicuna-7B | 85.6 | 24.0 |
| LLaVA-1.5 + **MoDA** | Vicuna-7B | **87.1** | **36.0** |
| LLaVA-MoRE OpenAI CLIP (Cocchi et al., 2025) | LLaMA-8B | 85.1 | 27.3 |
| LLaVA-MoRE OpenAI CLIP + **MoDA** | LLaMA-8B | **86.3** | **28.7** |
| LLaVA-MoRE SigLIP-S2 (Cocchi et al., 2025) | LLaMA-8B | 86.0 | 39.3 |
| LLaVA-MoRE SigLIP-S2 + **MoDA** | LLaMA-8B | **87.7** | **42.7** |
| Qwen3-VL-2B-Instruct (Qwen Team, 2025) | Qwen3-2B | 89.4 | - |
| Qwen3-VL-2B-Instruct + **MoDA** | Qwen3-2B | **89.9** | - |

**Vision Centric Tasks.** On the benchmarks that require careful visual discrimination, shown in Table 2, architectural precision outperforms parameter count and follows our motivation. Patch tokenization mixes multiple semantics inside each token. MoDA applies cross-attentive, instruction-conditioned channel modulation that separates useful signals from unrelated content and routes them more effectively to the decoder. This converts the representational headroom in stronger encoders into measurable accuracy. Within the LLaVA-MoRE family, OpenAI CLIP with MoDA reaches the best LLaVA-Wild score at 73.9, and SigLIP-S2 with MoDA attains the strongest LLaVA-family results on MM-Star at 38.5, on V*Bench at 44.8, and on CV-Bench at 62.2. Within the LLaVA-1.5 family, the peak on MM-Vet is achieved by the compact Vicuna-7B with MoDA at 29.9. These datasets emphasize different skills such as recognition, reading, and spatial reasoning, yet the pattern is consistent. The largest gains appear when MoDA is paired

with SigLIP-S2, which provides richer features that MoDA can selectively emphasize. Importantly, MoDA also competes with models trained on larger and different data distributions. ShareGPT-4V, reported in gray, records 72.6 on LLaVA-Wild, 33.0 on MMStar, and 61.8 on CV-Bench. MoDA surpasses these results with 73.9 on LLaVA-Wild, 38.5 on MMStar, and 62.2 on CV-Bench. Comparisons to 13B baselines, including ShareGPT-4V, indicate that an 8B-class model with MoDA can meet or exceed larger systems where direct comparisons exist. This favors design choices that direct information flow over simply adding parameters and matches the behavior predicted by the method.

**Hallucination Detection.** MoDA's design intent is most evident on hallucination benchmarks, as summarized in Table 3. By emphasizing instruction-relevant channels and attenuating distractors, the model reduces reliance on priors and keeps outputs consistent with the visible content. With Vicuna-7B, MMVP improves by 12.0 points, from 24.0 to 36.0. With SigLIP-S2, MoDA attains the strongest LLaVA-MoRE scores on both tasks, reaching 87.7 on POPE and 42.7 on MMVP, and surpasses the 13B LLaVA 1.5 baseline, which records 85.9 on POPE and 24.7 on MMVP. Taken together, the results confirm three discoveries. First, MoDA scales with stronger encoders, most clearly with SigLIP-S2. Second, architectural refinement yields larger benefits than parameter growth in multiple settings. Third, hallucination detection is where MoDA delivers its most decisive gains for CLIP-family encoders. Across all three categories, MoDA delivers consistent improvements over each of the three MLLM architectures we evaluate. These gains are consistent with the mechanism described in Section 3, where instruction-conditioned channel modulation reduces the influence of mixed patch semantics. The improvements require no additional supervision or changes to the training protocol, indicating that MoDA improves how existing evidence is used rather than expanding data or labels.

**Generalization to Non-CLIP Encoders.** To verify that MoDA's gains extend beyond the CLIP-family encoders used in LLaVA-1.5 and LLaVA-MoRE, we integrate the same module into Qwen3-VL-2B-Instruct (Qwen Team, 2025), a recent state-of-the-art MLLM whose native ViT departs from the CLIP backbone. Despite this architectural shift and the substantially stronger baseline, MoDA continues to deliver consistent gains across visual question answering: GQA improves by 3.8 points (from 59.4 to 63.2), ScienceQA by 4.9 points (from 79.3 to 84.2), RealWorldQA by 4.1 points (from 64.7 to 68.8), and MMBench by 0.9 points (from 86.5 to 87.4). Vision-centric tasks improve as well: MMStar rises from 53.9 to 55.3, CV-Bench from 80.9 to 81.0, and POPE from 89.4 to 89.9. ChartQA regresses from 80.0 to 79.0 and V*Bench from 77.0 to 74.9, consistent with the mechanism described in Section 3: MoDA

contributes most when the encoder's representations under-utilize task-relevant channels, while encoders already specialized for chart reading or high-resolution visual search leave less headroom for channel-wise refinement. These cross-architecture results confirm that instruction-guided channel modulation generalizes beyond CLIP, validating the mechanism as a general post-alignment refinement rather than a CLIP-specific remedy.

## 4.2. Ablation Studies

We conduct systematic ablations to address three key questions: **(i)** Why Cross-Attention outperforms linear modulation, **(ii)** Whether improvements stem from architecture vs. added capacity, **(iii)** Component synergy effects across different encoders and LLMs.

**Cross-Attention vs. Alternatives:** To understand why Cross-Attention outperforms alternatives, we analyze how each approach handles queries requiring disentangling mixed visual semantics within individual patches. The three approaches differ fundamentally: Linear MLP applies the same transformation regardless of instruction, Self-Attention concatenates features without explicit cross-modal conditioning, while Cross-Attention uses visual features as queries and instruction tokens as memory, enabling selective channel emphasis based on instruction semantics. This architectural difference becomes crucial when processing patches containing multiple semantic elements, as Cross-Attention can dynamically weight channels corresponding to instruction-relevant concepts while suppressing irrelevant information. With SigLIP-S2, Cross-Attention achieves the highest performance (69.4 vs Self-Attention 68.0 vs Linear 67.3) with substantial gains on reasoning tasks: ScienceQA shows Cross-Attention at 81.9 compared to Self-Attention 79.9 and Linear 77.8, while MMVP demonstrates Cross-Attention's 42.7 versus Self-Attention's 39.5 and Linear's 40.0.

**Architecture vs. Capacity:** The performance patterns argue against pure capacity effects: task-specific rather than uniform improvements (MMVP shows large gains while other tasks show smaller improvements), consistent improvement patterns across different LLM backbones, and architectural choice matters more with stronger components (differences are minimal with CLIP but substantial with SigLIP-S2).

**Component Synergy:** $L_1$ regularization consistently degrades Cross-Attention performance across both encoders, while Linear MLP remains largely unaffected. The degradation is particularly severe for fine-grained reasoning. LLaMA 3.1-8B provides modest improvements over Vicuna-7B, while SigLIP-S2 dramatically amplifies MoDA's effectiveness (+5.1 points over CLIP), confirming that instruction-guided modulation becomes increasingly valuable with

*Table 4.* **Ablation Study of MoDA Components.** We systematically evaluate MoDA architecture variants (Linear MLP vs. Cross-Attention vs. Self-Attention), auxiliary supervision ($L_1$ vs. None), LLM backbones (Vicuna-7B vs. LLaMA 3.1-8B), and vision encoders (CLIP vs. SigLIP-S2). Cross-Attention without auxiliary loss consistently outperforms alternatives, with benefits amplified by stronger visual encoders. Bold values indicate best performance per column.

| MoDA Type | Supp. Loss | LLM | Vision Encoder | POPE | GQA | SQA | MMVP | Avg. |
|---|---|---|---|---|---|---|---|---|
| *Baseline Models (No MoDA)* | | | | | | | | |
| - | - | Vicuna-7B | CLIP ViT-L/14@336 | 85.6 | 62.4 | 69.0 | 24.0 | 60.3 |
| - | - | LLaMA 3.1-8B | CLIP ViT-L/14@336 | 85.1 | 63.6 | 76.3 | 27.3 | 63.1 |
| - | - | LLaMA 3.1-8B | SigLIP-S2 | 86.0 | 64.9 | 77.1 | 39.3 | 66.8 |
| *CLIP ViT-L/14@336 Ablations* | | | | | | | | |
| Linear (MLP) | $L_1$ | LLaMA 3.1-8B | CLIP ViT-L/14@336 | 87.2 | 64.3 | 76.7 | 28.7 | 64.2 |
| Linear (MLP) | None | LLaMA 3.1-8B | CLIP ViT-L/14@336 | 86.6 | 64.4 | 77.8 | 28.1 | 64.2 |
| Cross-Attention | $L_1$ | LLaMA 3.1-8B | CLIP ViT-L/14@336 | 87.6 | 64.2 | 76.8 | 20.2 | 62.2 |
| Self-Attention | None | LLaMA 3.1-8B | CLIP ViT-L/14@336 | 86.5 | 64.2 | 77.3 | 27.9 | 64.0 |
| Cross-Attention | None | LLaMA 3.1-8B | CLIP ViT-L/14@336 | 86.3 | 64.4 | 77.8 | 28.7 | 64.3 |
| *LLM Backbone Comparison* | | | | | | | | |
| Cross-Attention | None | Vicuna-7B | CLIP ViT-L/14@336 | 87.1 | 62.5 | 71.0 | 36.0 | 64.2 |
| *SigLIP-S2 Ablations* | | | | | | | | |
| Linear (MLP) | $L_1$ | LLaMA 3.1-8B | SigLIP-S2 | 85.8 | 65.2 | 77.9 | 39.6 | 67.1 |
| Linear (MLP) | None | LLaMA 3.1-8B | SigLIP-S2 | 86.6 | 64.8 | 77.8 | 40.0 | 67.3 |
| Cross-Attention | $L_1$ | LLaMA 3.1-8B | SigLIP-S2 | 87.0 | 65.1 | 79.2 | 41.1 | 68.1 |
| Self-Attention | None | LLaMA 3.1-8B | SigLIP-S2 | **87.9** | 64.9 | 79.9 | 39.5 | 68.0 |
| Cross-Attention | None | LLaMA 3.1-8B | SigLIP-S2 | 87.7 | **65.4** | **81.9** | **42.7** | **69.4** |

richer visual representations.

**Additional Analysis.** We provide extended analysis in the Appendix. For ablation studies, we examine the effect of MoDA adapter depth with Linear MLP (Table S2) and Cross-Attention layers (Table S3) in Section A.3.1, as well as the impact of MoDA placement within the LLM (Table S4) in Section A.3.2. For interpretability, we provide attention map visualizations (Figure S1) in Section A.4 that demonstrate how MoDA concentrates attention on task-relevant regions. Finally, Section A.5 presents qualitative examples (Figure S2) comparing MoDA against baselines on challenging MMVP instances, showing improved fine-grained grounding and format compliance.

*Table 5.* **Comparison with masking approaches.** MoDA vs. token-level masking (LLaMA 3.1-8B + SigLIP-S2). **Bold**: best per column.

| Strategy | POPE | GQA | SQA | MMVP | Avg |
|---|---|---|---|---|---|
| Baseline | 86.0 | 64.9 | 77.1 | 39.3 | 66.8 |
| Learn. Mask (Barrios & Jin, 2024) | 86.9 | 65.1 | 79.9 | 41.9 | 68.5 |
| Sparse Mask (Lin et al., 2022) | 85.8 | 64.7 | 76.7 | 38.8 | 66.5 |
| **MoDA (ours)** | **87.7** | **65.4** | **81.9** | **42.7** | **69.4** |

### 4.3. Comparison with Masking Approaches

Table 5 validates our core hypothesis by comparing MoDA against token-level masking methods under identical conditions. MoDA achieves the highest performance across all

benchmarks, establishing clear superiority with 69.4 average performance compared to 68.5 for Learnable Masking and 66.5 for Sparse Masking. Most importantly, MoDA reaches the strongest results on fine-grained tasks: 42.7 on MMVP and 81.9 on ScienceQA. While token-level masking operates on discrete attention weights and requires layer-wise computation scaling with model depth, MoDA's channel-wise modulation provides continuous, instruction-guided refinement with single-pass efficiency, enabling more effective visual-language understanding without computational overhead that increases linearly with the number of transformer layers.

*Table 6.* **Computational overhead of MoDA relative to LLaVA-MoRE.** MoDA introduces minimal overhead with only 3.7% of total parameters and less than 1% of computational operations (MACs and FLOPs), showing that performance gains stem from architectural innovation rather than scaling.

| Metric | MoDA | LLaVA-MoRE (8B) | Ratio (%) |
|---|---|---|---|
| Parameters | 0.302B | 8.0B | 3.7 |
| MACs | 45.1G | ≈ 5,246G | 0.86 |
| FLOPs | 90.2G | ≈ 10,492G | 0.86 |

### 4.4. Computational Efficiency Analysis

MoDA introduces minimal overhead, adding only 3.7% parameters and <1% MACs/FLOPs compared to LLaVA-MoRE (8B) (Table 6), confirming gains stem from architectural design rather than scaling. MoDA's strategic placement

after the adapter and before the LLM enables instruction-guided modulation with optimal efficiency-performance tradeoffs, as validated by our ablation studies comparing different placement strategies (Appendix A.3.2). This positioning allows MoDA to operate on pre-aligned visual features while maintaining computational efficiency. In multi-turn scenarios, visual features are cached once, with subsequent queries requiring only modulation recomputation ($<1\%$ computation).

## 5. Limitations

MoDA works by directly modulating the channels in the input, but it cannot achieve explicit sparsity in the channel dimension. That is, MoDA re-weights the channel dimension but only occasionally sets a channel's weight to zero. Such a property could be desirable to make stronger feature selection and effectively guide the attention of the LLM towards more semantically relevant features. Furthermore, MoDA's benefit depends on whether the encoder's representations underutilize task-relevant channels. On Qwen3-VL we observe small regressions on ChartQA and V*Bench, where the vision encoder is already specialized for chart reading and high-resolution visual search, leaving limited headroom for channel-wise refinement (see Section 4). Additionally, our evaluation focuses on single-image benchmarks; extending MoDA to multi-image and video understanding remains future work.

## 6. Conclusions

We have introduced MoDA, a novel modulation adapter for MLLMs that works as an ad-hoc module. At its core, MoDA re-weights the contribution of each individual visual feature channel based on the early language embeddings of the language prompt. The re-weighted set of features acts as an implicit feature selector, promoting the visual features that are most relevant for each individual query, thus improving the performance of MLLMs. Across multiple benchmarks and multiple MLLM architectures MoDA shows consistent performance improvements over the baselines. MoDA does not require any additional pre-training or supervision. By simply appending MoDA to the MLLM during the instructional tuning phase, we observe direct improvements across diverse benchmarks.

Several directions remain open for future work. First, MoDA currently performs soft channel re-weighting rather than explicit channel selection; incorporating sparsity-inducing objectives that drive uninformative channels to zero could yield stronger and more interpretable feature selection. Second, our analysis shows that MoDA's benefit scales with the degree to which an encoder underutilizes task-relevant channels, suggesting that pairing MoDA with

backbones that are not already specialized for a target domain is most promising; characterizing this relationship across a wider range of visual encoders is a natural next step. Finally, while our evaluation centers on single-image understanding, the channel-modulation mechanism is agnostic to the number of inputs, and extending MoDA to multi-image reasoning and video understanding is a promising avenue for instruction-guided visual refinement at scale.

## Impact Statement

This paper introduces MoDA, a lightweight module that improves fine-grained visual understanding in Multimodal Large Language Models by reducing hallucinations and enhancing visual grounding. We believe this work can have positive societal impact across several domains.

**Positive Applications.** More reliable visual understanding can benefit accessibility tools for visually impaired users, where accurate image descriptions are essential. Educational applications can leverage improved visual reasoning for science tutoring and diagram interpretation. Medical imaging analysis may benefit from reduced hallucinations when describing radiological findings, though clinical deployment would require extensive validation beyond our benchmarks.

**Computational Accessibility.** MoDA adds less than 1% computational overhead and 3.7% parameters, enabling performance improvements without requiring substantially larger models or infrastructure. This efficiency may help democratize access to more reliable vision-language systems for researchers and practitioners with limited computational resources.

**Potential Risks.** As with all advances in vision-language models, improved capabilities could be misused for generating misleading visual descriptions or manipulating image-based content. We encourage responsible deployment with appropriate safeguards and content verification mechanisms.

**Data and Privacy.** Our work does not introduce new data collection practices. We train and evaluate exclusively on publicly available benchmarks using standard protocols, and we do not use any private or sensitive imagery.

## Acknowledgments

This work was supported by startup funds provided by Dartmouth College. In addition, the research reported in this publication was supported by funding from King Abdullah University of Science and Technology (KAUST) - Center of Excellence for Generative AI, under award number 5940.

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

# A. Appendix

## A.1. Experiment Setup

**Visual Question Answering Benchmarks.** These benchmarks evaluate models' capability to accurately answer questions requiring visual reasoning and comprehension. **GQA** (Hudson & Manning, 2019) builds upon Visual Genome's scene graph annotations and contains 113k images with 22 million questions emphasizing compositional reasoning and scene understanding. **ScienceQA** (Lu et al., 2022) assesses models using complex multimodal multiple-choice questions spanning three major subject areas (natural science, language science, and social science), encompassing 26 topics, 127 categories, and 379 distinct skills across 4,241 test examples. **MMBench** (Liu et al., 2023b) consists of approximately 3,000 multiple-choice questions spanning 20 diverse domains, designed to rigorously assess MLLM capabilities across perception and reasoning paradigms through a structured hierarchical taxonomy. **RealWorldQA** (xAI, 2024) contains over 700 real-world images captured from vehicles and other scenarios, each paired with spatial reasoning questions that evaluate real-environment understanding and physical scene comprehension. **ChartQA** (Masry et al., 2022) focuses on chart understanding with 9.6k human-written and 23k auto-generated questions across approximately 20k charts (bar, line, pie), requiring visual and logical reasoning such as comparing values, identifying trends, and performing arithmetic operations over chart data.

**Vision-Centric Benchmarks.** These benchmarks specifically target fine-grained visual understanding and detailed image analysis capabilities. **LLaVA-Wild** (Liu et al., 2023a) comprises 24 diverse images with 60 questions spanning indoor and outdoor scenes, memes, paintings, and sketches, with each image paired with detailed, manually curated descriptions and targeted questions categorized into conversation, detailed description, and complex reasoning. **MM-Vet** (Yu et al., 2024) includes 200 test images with 218 questions covering six core vision-language capabilities: recognition, knowledge, optical character recognition (OCR), spatial awareness, language generation, and mathematics, often requiring integration of multiple skills for accurate responses. **MMStar** (Chen et al., 2024b) presents 1,500 manually curated multimodal challenge items with minimal overlap, evaluating six high-level capabilities across 18 fine-grained axes and targeting complex visual dependency and reasoning tasks where visual content is essential for answering. **V*Bench** (Zhang et al., 2024) evaluates detailed visual analysis using 191 high-resolution images from SAM with average resolution of 2246×1582, containing two sub-tasks: attribute recognition (115 samples requiring recognition of object attributes like color and material) and spatial relationship reasoning (76 samples asking for relative spatial relationships between objects). **CV-Bench** (Tong et al., 2024a) provides a comprehensive evaluation framework with 2,638 manually-inspected examples, repurposing standard vision benchmarks such as ADE20K, COCO, and Omni3D to assess both 2D understanding (spatial relationships, object counting) and 3D understanding (depth order, relative distance) within a multimodal context.

**Hallucination Detection Benchmarks.** These benchmarks specifically measure model tendency to generate false or

inconsistent information not present in the visual input. **POPE** (Li et al., 2023b) evaluates object hallucination through 8,910 binary classification queries across three sub-sets (random, popular, and adversarial), each constructed via distinct sampling strategies to probe different aspects of hallucination phenomena in MLLMs. Following standard practice, we report average performance across all three subsets. **MMVP** (Tong et al., 2024b) measures hallucination through 150 carefully constructed image pairs, each accompanied by two binary-choice questions. The image pairs are selected to have highly similar CLIP embeddings, and accurate performance requires both questions per pair to be answered correctly, making this benchmark particularly challenging for detecting subtle visual differences and avoiding spurious correlations.

## A.2. Implementation Details

Table S1 summarizes all optimization, hardware, and architectural specifications needed to reproduce our results. We followed LLaVA's (Liu et al., 2023a; 2024) established two-stage training curriculum. First, we train the adapters on 558K alt-text image-caption pairs, then fine-tune the network on high-quality visual instruction data. Both stages optimize the same next-token prediction objective, allowing us to maintain optimizer state and the cosine learning rate schedule with 3% warm-up across stages.

*Table S1.* **Training Configuration Summary.** This table details the training and fine-tuning hyper-parameters. "PT" refers to the pre-training stage using large-scale alt-text image–caption data, while "FT" denotes the fine-tuning stage on high-quality visual-instruction datasets.

| Hyper-parameter | PT | FT |
|---|---|---|
| Global batch size | 256 | 128 |
| Effective epochs | 1 | 1 |
| Learning rate | $1 \times 10^{-3}$ | $2 \times 10^{-5}$ |
| LR schedule | Cosine decay with 3 % warm-up | |
| Weight decay | 0 | |
| Optimiser | AdamW | |
| DeepSpeed stage | 2 | 3 |
| *Hardware* | | |
| GPU type | A100/H100 (80 GB) | |
| Deployment | Multi-node cluster | |
| *Model components (shared across stages)* | | |
| Language backbone | LLaMA-3.1-8B or Vicuna-7B | |
| Visual encoder | CLIP or SigLIP with S2 multiscale | |
| Adapter (MoDA) | 2 × cross-attention; 16 heads | |

To ensure direct comparability, we matched all hyper-parameters (batch sizes, learning rates, weight decay, and optimizer choice) exactly as reported in LLaVA-1.5. Training was distributed across multi-node clusters using 80 GB

A100 or H100 GPUs with DeepSpeed ZeRO Stage-2 for pre-training and Stage-3 for fine-tuning. The model architecture combines either LLaMA-3.1-8B (Llama Team, AI @ Meta, 2024) or Vicuna-7B (Zheng et al., 2023) as the language backbone with CLIP (Radford et al., 2021) or SigLIP (Zhai et al., 2023) image encoders. Visual and textual information merge through a two-layer MoDA cross-attention block where visual tokens query instruction embeddings.

**MMVP Benchmark.** To evaluate performance on the MMVP benchmark, we opted for an open-source and cost-effective alternative to proprietary language models. Instead of using GPT-4, we employed Gemma 3 (Team, 2025) as the grader (using only text). This model was deployed using `Ollama`, which ensures compatibility with the OpenAI API. This setup allowed us to maintain seamless integration with our Python-based evaluation pipeline while significantly reducing operation costs without compromising evaluation consistency.

## A.3. Additional Ablation Studies

### A.3.1. EFFECT OF MODA ADAPTER DEPTH

**Linear (MLP) Depth.** Table S2 showcases the impact of the adapter depth across four different evaluation protocols: POPE and MMVP, which target hallucination robustness; ScienceQA (SQA), which probes scientific reasoning; and GQA, a dataset for real world visual reasoning and compositional question answering. Note that the first row mirrors line 5 of Table 2 in the main paper. When we increase the MLP depth from two to four layers the average score falls by nearly twelve points with the largest drops on GQA and ScienceQA, suggesting that extra layers hinder the model's ability to align visual evidence with language semantics. We also do not observe any improvement in hallucination tests using POPE and MMVP. This indicates that deeper adapters add complexity without strengthening actual grounding.

**Cross-Attention Depth.** Table S3 extends our analysis by systematically varying the number of Cross-Attention layers in MoDA. The results confirm that performance remains stable across 2, 3, and 4 layers (64.2–64.3 average), with no statistically significant differences. This validates our selection of the 2-layer configuration as it achieves equivalent accuracy with minimal computational overhead. Unlike the Linear (MLP) variant which degrades significantly with increased depth, Cross-Attention maintains consistent performance, suggesting that the cross-modal interaction mechanism is more robust to depth variations.

**Takeaway.** With the current data regime, increasing depth does not improve understanding, and MoDA with two Cross-Attention layers remains the clear choice for balancing multimodal alignment, reasoning precision, and resistance to hallucination.

*Table S2.* **Ablation on MoDA Depth (Linear MLP).** Effect of increasing the number of layers in the MoDA adapter while keeping every other component fixed. Scores are reported on POPE, GQA, SQA and MMVP; the final column shows the mean across tasks.

| MoDA type | # layers | Supp. loss | LLM | Vision enc. | POPE | GQA | SQA | MMVP | Avg |
|---|---|---|---|---|---|---|---|---|---|
| Linear (MLP) | 2 | None | LLaMA 3.1-8B | CLIP ViT-L/14@336 | **86.6** | **64.4** | **77.8** | **28.1** | **64.2** |
| Linear (MLP) | 4 | None | LLaMA 3.1-8B | CLIP ViT-L/14@336 | 82.0 | 57.7 | 42.1 | 27.3 | 52.3 |

*Table S3.* **Ablation on Cross-Attention Depth.** Effect of varying the number of Cross-Attention layers in MoDA while keeping all other components fixed. Performance remains stable across 2, 3, and 4 layers, validating our selection of 2 layers as the default configuration.

| MoDA type | # layers | Supp. loss | LLM | Vision enc. | POPE | GQA | SQA | MMVP | Avg |
|---|---|---|---|---|---|---|---|---|---|
| Cross-Attention | 2 | None | LLaMA 3.1-8B | CLIP ViT-L/14@336 | 86.3 | **64.4** | 77.8 | 28.7 | 64.3 |
| Cross-Attention | 3 | None | LLaMA 3.1-8B | CLIP ViT-L/14@336 | **86.4** | 63.8 | **78.2** | **29.0** | **64.3** |
| Cross-Attention | 4 | None | LLaMA 3.1-8B | CLIP ViT-L/14@336 | 86.2 | 64.0 | 77.9 | 28.8 | 64.2 |

*Table S4.* **Impact of MoDA placement in the LLM.** Comparison of LLaVA-MoRE 8B without MoDA, with MoDA injected at the beginning of the LLM module, and with MoDA applied to every block of the LLM module. Scores are reported on POPE and MMVP (hallucination robustness), ScienceQA (scientific reasoning), and GQA (real-world visual reasoning); the final column shows the mean across tasks.

| Model | LLM size | LLM | MoDA position | Vision enc. | POPE | GQA | SQA | MMVP | Avg |
|---|---|---|---|---|---|---|---|---|---|
| LLaVA-MoRE 8B | 8B | LLaMA 3.1-8B | - | SigLIP-S2 | 86.0 | 64.9 | 77.1 | 39.3 | 66.8 |
| LLaVA-MoRE 8B + MoDA | 8B | LLaMA 3.1-8B | All layers in LLM | SigLIP-S2 | 86.3 | 65.1 | 78.9 | 39.8 | 67.5 |
| LLaVA-MoRE 8B + MoDA | 8B | LLaMA 3.1-8B | Beginning | SigLIP-S2 | **87.7** | **65.4** | **81.9** | **42.7** | **69.4** |

### A.3.2. EFFECT OF MODA PLACEMENT IN THE LLM

Table S4 shows that applying MoDA more broadly within the LLM does not necessarily improve performance. Injecting MoDA only at the beginning of the LLM increases the average score from 66.8 to 69.4 (+2.6). In contrast, applying MoDA to every transformer block yields a smaller improvement of +0.7, raising the mean to 67.5.

Across individual benchmarks, the beginning-only configuration achieves the largest gains: +1.7 on POPE, +0.5 on GQA, +4.8 on ScienceQA, and +3.4 on MMVP relative to the baseline. The all-layers variant does not surpass these improvements on any task. This difference is further amplified by computational cost: applying MoDA to all transformer blocks increases training time from roughly 20 hours to more than 50 hours, whereas injecting MoDA only at the beginning preserves the original training budget.

**Takeaway.** MoDA is most effective when injected at the beginning of the LLM, offering the best trade-off between accuracy and efficiency. This setting improves the mean from 66.8 to 69.4 (+2.6) while maintaining a $\sim 20$ hour training budget. Distributing MoDA across all LLM layers provides only a +0.7 gain but extends training beyond 50 hours. Therefore, we adopt beginning-only MoDA as the default, and reserve all-layers MoDA for cases where marginal gains justify substantially higher compute.

### A.4. Attention Map Visualization

To provide insight into how MoDA improves visual grounding, we visualize attention maps derived from the LLM's self-attention layers. In MLLMs, visual tokens are concatenated with language tokens and processed jointly through the transformer layers. We extract the attention weights from the output token positions attending to the visual token positions, then spatially reshape these weights to match the original image resolution. Figure S1 presents a representative example from ScienceQA, where the task requires comparing the average kinetic energies of gas particles across two containers to determine which sample exhibits higher temperature. The baseline model produces a diffuse attention distribution across both containers and irrelevant background regions, indicating that the LLM struggles to focus on task-relevant visual tokens, leading to an incorrect prediction. In contrast, when visual features are preprocessed through MoDA's channel-wise modulation, the attention maps exhibit concentrated activation patterns localized on Sample A's particles and their associated velocity indicators, which are directly relevant to solving the task. This demonstrates that MoDA's instruction-guided modulation effectively refines visual token representations, enabling the LLM to allocate attention more precisely to task-relevant regions. These visualizations provide interpretable evidence that channel-wise feature modulation enhances visual-language alignment, facilitating accurate fine-grained

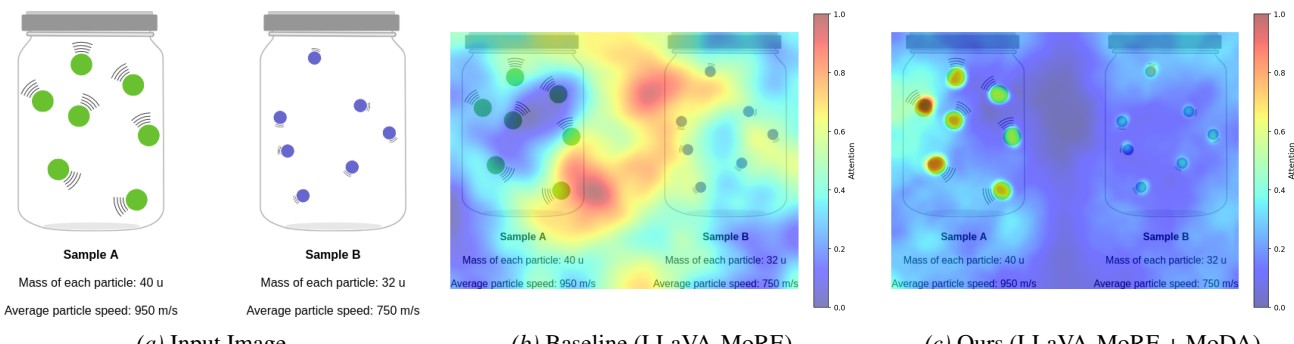

*(a)* Input Image      *(b)* Baseline (LLaVA-MoRE)      *(c)* Ours (LLaVA-MoRE + MoDA)

*Figure S1.* **Attention map visualization on ScienceQA.** Given the question *"Which sample has the higher temperature?"*, the baseline model (b) exhibits diffuse attention across both containers and irrelevant regions, leading to an incorrect response. In contrast, MoDA (c) concentrates attention on Sample A's particles and motion indicators, enabling the model to correctly identify Sample A as having higher temperature.

visual reasoning.

## A.5. Qualitative Analysis

Figure S2 presents a qualitative comparison between the baseline LLaVA-MoRE using SigLIP-S2 (denoted as LM) and our proposed MoDA, which augments the same baseline with a modulation adapter to enhance visual representation quality. The first two examples involve determining whether the letter *D* is present in a keyboard layout. In the first case, LM incorrectly identifies the presence of the letter *D* despite its absence in the image and fails to select a valid multiple-choice option, resulting in both an incorrect response and invalid output format. In the second case, LM correctly identifies the letter's presence and selects the appropriate answer. In contrast, MoDA consistently selects the correct alternatives: "(a) Correct" for the first example and "(b) Incorrect" for the second, demonstrating its ability to produce concise outputs that comply with the required format while maintaining fine-grained visual understanding.

In the third example, both the baseline and MoDA produce correct answers but fail to follow the multiple-choice format. The fourth case involves identifying whether a snake's tongue is touching its skin, a subtle perceptual task where the correct answer is "No". LM misclassifies this contact while MoDA provides the correct answer, demonstrating greater sensitivity to localized visual features (fine-grained details).

The fifth and sixth examples test whether textual markings are present on a police van's light bar. In the fifth case, both models provide correct answers, but only MoDA follows the required output format instructions. In the sixth example, the LM incorrectly predicts the presence of text, likely due to overgeneralized visual priors, which are assumptions (hallucinations) formed from pre-training data that cause the model to expect text in similar visual contexts even when none is present. In contrast, MoDA accurately iden-

tifies the absence of text and maintains proper formatting. These results highlight MoDA's improved grounding in visual evidence and its stronger compliance with formatting requirements, closely following the user's instructions.

## A.6. The Use of Large Language Models (LLMs)

We used commercial large language models (e.g., ChatGPT) only as editorial tools to improve the manuscript's readability. Their role was limited to language editing, such as correcting grammar, improving clarity, and smoothing the flow of text, and they did not influence the research design, data analysis, or the research conclusions.

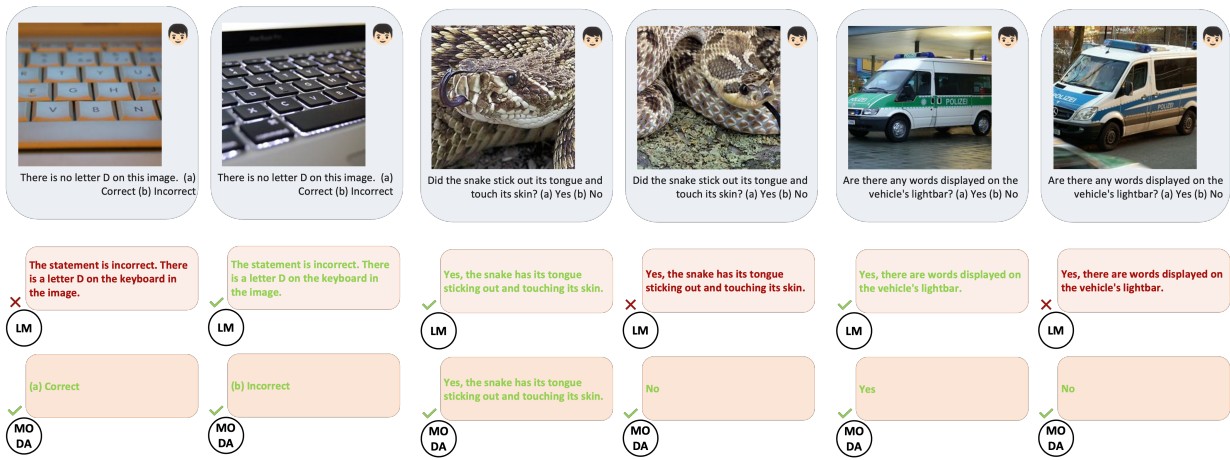

*Figure S2.* **Qualitative Analysis.** Qualitative comparison between the baseline LLaVA-More SigLIP-S2 (denoted **LM**) and LLaVA-More SigLIP-S2 + MoDA (denoted **MoDA**). Each column shows a multiple-choice VQA instance from the MMVP benchmark. ✗ marks an incorrect prediction, whereas ✔ denotes a correct one. Although the baseline frequently produces lengthy free-form answers that do not match the question format, MoDA consistently selects the correct alternative, successfully addressing the task. From left to right, we observe: (**i** & **ii**) recognition of a specific keyboard key, (**iii** & **iv**) detection of subtle tongue–skin contact in a snake, and (**iv** & **v**) identification of printed text on a police vehicle's light bar. Across all examples, MoDA demonstrates superior fine-grained grounding of visual cues.

