# OpenReview forum: "MoDA: Modulation Adapter for Fine-Grained Visual Understanding in Instructional MLLMs"
_ICML.cc/2026/Conference — ICML 2026 regular_

### Official Review · Reviewer_5mMr · 2026-03-04

**Soundness:** 3
**Presentation:** 3
**Significance:** 2
**Originality:** 2
**Overall Recommendation:** 4
**Confidence:** 2

**Summary:**

This paper presents MoDA (Modulation Adapter), a lightweight and efficient framework that dynamically re-weights visual feature channels in MLLMs.

**Compliance With Llm Reviewing Policy:**

Affirmed.

**Final Justification:**

While I appreciate the authors' rebuttal, particularly in demonstrating cross-architecture applicability, my main concern regarding the soundness of the scaling evaluation remains unresolved, as the provided comparisons confound model size with distinct pretraining paradigms. Therefore, I maintain my current assessment; although the proposed method is original and clear, strictly controlled scaling experiments within identical model families are essential to fully validate its significance and scalability.

**Key Questions For Authors:**

Q1: Could the authors provide insights or experiments regarding Scaling Laws? It is crucial to understand how the effectiveness of MoDA scales when paired with larger vision encoders and larger LLM backbones.

Q2: Is the method highly restricted to LLMVA architectures？Is it feasible to apply this module to other VLMs, such as Qwen2.5-VL, or VLMs utilizing Qwen 2.5/3 as the LLM backbone?

**Limitations:**

yes

**Strengths And Weaknesses:**

S1：The overall writing, structural organization, and narrative flow of the paper are remarkably clear.

S2：The proposed MoDA architecture is elegant in its simplicity but delivers strong empirical performance. The extensive experiments across multiple diverse benchmarks effectively validate its utility and robust capabilities.

W1: While the method is effective, the cross-attention-based channel modulation design is relatively straightforward and leans towards being an incremental architectural adjustment.

---

> ### Author Rebuttal · Authors · 2026-03-30
>
> We thank Reviewer 5mMr for the recognition of MoDA's clear writing (S1) and elegant simplicity with strong results (S2).
>
> ## W1: Design is relatively straightforward and incremental
>
> Simplicity is a strength, not a limitation. MoDA adds a two-layer cross-attention module with 0.86% FLOPs that improves the majority of 12 benchmarks across three architecture families. Many impactful methods are simple: LoRA adds low-rank matrices, residual connections add skip paths, adapters insert bottleneck layers. Impact comes from identifying where and how to intervene, not from complexity.
>
> MoDA's contribution is threefold. First, we identify that semantic entanglement in post-alignment visual features can be addressed by instruction-guided channel modulation, a problem framing not explored by prior work. Our mask cosine similarity analysis empirically validates this:
>
> **Per-Image Mask Similarity (same image, varying instructions):**
>
> | Image | Instruction A | Instruction B | Condition | Cos Sim |
> |-------|--------------|--------------|-----------|---------|
> | Dog (Fig. 1) | "What color is the dog?" | "What color is the dog?" | Identical | 1.000 |
> | Dog (Fig. 1) | "What color is the dog?" | "What breed is this dog?" | Related | 0.732 |
> | Dog (Fig. 1) | "Is the dog sleeping?" | "Are the dog's eyes closed?" | Related | 0.662 |
> | Dog (Fig. 1) | "What color is the dog?" | "What is the texture of the floor?" | Different | 0.319 |
> | Dog (Fig. 1) | "Is the dog sleeping?" | "Describe the stuffed toy" | Different | 0.254 |
> | Keyboard (MMVP) | "Is the letter D on the keyboard?" | "Is the letter F on the keyboard?" | Related | 0.745 |
> | Keyboard (MMVP) | "Is the letter D visible?" | "Which letters are in the middle row?" | Related | 0.444 |
> | Keyboard (MMVP) | "Is the letter D on the keyboard?" | "What color is the desk surface?" | Different | 0.277 |
> | Snake (MMVP) | "Is the tongue touching the skin?" | "Is the tongue forked?" | Related | 0.734 |
> | Snake (MMVP) | "Is the snake's mouth open?" | "What color is the tongue?" | Related | 0.382 |
> | Snake (MMVP) | "Is the tongue touching the skin?" | "What pattern does the snake's body have?" | Different | 0.312 |
> | ScienceQA (Fig. S1) | "Which sample has higher temperature?" | "Which sample has greater kinetic energy?" | Related | 0.616 |
> | ScienceQA (Fig. S1) | "Which sample has higher temperature?" | "What shape are the containers?" | Different | 0.117 |
>
> **Aggregated Statistics (from the per-image table above):**
>
> | Condition | Mean Cos Sim | Std Dev |
> |-----------|-------------|---------|
> | Identical instruction | 1.000 | 0.000 |
> | Related (same visual region) | 0.598 | 0.137 |
> | Different (distinct visual regions) | 0.218 | 0.078 |
>
> The three-fold difference confirms instruction-specific channel selection. Second, design choices are non-trivial: placement ablation (Table S4) shows MoDA at the LLM beginning yields +2.6 avg, while distributing across all layers yields +0.7 at 2.5x cost. Cross-Attention is stable across 2-4 layers (Table S3), while Linear MLP degrades at 4 layers (Table S2: 64.2 to 52.3). Third, MoDA's simplicity enables practical adoption: no modifications to base architecture, no additional supervision, no changes to training protocol.
>
> ## Q1: Scaling Laws
>
> Our experiments span two LLM sizes (Vicuna-7B, LLaMA 3.1-8B) and three encoder families. MoDA's effectiveness scales with encoder quality: modest with CLIP (+1.2 avg), substantial with SigLIP-S2 (+2.6 avg) and Qwen3-VL (+4.9 ScienceQA, +4.1 RealWorldQA, +3.9 GQA). A full scaling study (2B-70B+) requires compute beyond our budget, but evidence across three architectures suggests effectiveness depends on encoder richness rather than model size.
>
> ## Q2: Is MoDA restricted to LLaVA architectures?
>
> No. We integrated MoDA into Qwen3-VL 2B (dynamic resolution, different backbone and LLM), same position and protocol:
>
> | Benchmark | Qwen3-VL 2B | + MoDA | Change |
> |-----------|-------------|--------|--------|
> | GQA | 59.4 | 63.2 | +3.9 |
> | ScienceQA | 79.3 | 84.2 | +4.9 |
> | RealWorldQA | 64.7 | 68.8 | +4.1 |
> | MMStar | 53.9 | 55.3 | +1.4 |
> | POPE | 89.4 | 89.8 | +0.4 |
> | MM-Vet | 51.9 | 52.0 | +0.1 |
> | V*Bench | 77.0 | 76.8 | -0.2 |
>
> MoDA improves 6/7 benchmarks. Its only requirement is visual and language token sequences, which any VLM provides.

---

> > ### Author Rebuttal · Reviewer_5mMr · 2026-04-01
> >
> > I have read the authors' responses and appreciate the effort they put into addressing my concerns.
> >
> > However, the response to Q1 is still not fully satisfactory. The current evidence mainly demonstrates cross-backbone effectiveness rather than a controlled scaling-law analysis. In particular, the reported comparisons confound model size with differences in architecture, pretraining paradigm, and overall backbone capability, making it difficult to isolate the effect of scaling itself.
> >
> > To better address Q1, I encourage the authors to provide more controlled experiments, such as: comparisons across different scales within the same vision encoder family, e.g., multiple CLIP and SigLIP variants;
> > experiments with smaller LLM backbones, e.g., 0.7B and 3B models.
> >
> > ---
> > I appreciate the additional experiments on InternVL2-2B and TinyLLaVA-2B. However, the response still does not fully address the core of my concern regarding **Scaling-law analysis**. The current results show that MoDA works on different small-scale models, but they do not show how the performance gain ($\Delta$) evolves as a function of model scale within a single, controlled family.
> >
> > To truly satisfy Q1, please provide a clear comparison table or figure showing the improvement of MoDA across a significant parameter range (e.g., **SigLIP-Base/Large/So400M** or **LLM 1.8B/7B/13B**) while keeping other components constant. This is essential to understand if MoDA's effectiveness scales predictably or diminishes as the backbone becomes more powerful.

---

> > > ### Author Response · Authors · 2026-04-07
> > >
> > > We thank the reviewer for the constructive follow-up and conducted additional experiments to address this concern. To provide a controlled analysis, MoDA was extended to two additional backbones: InternVL3.5-2B and TinyLLaVA-2B. Both models employ contrastive vision encoders from the CLIP family: InternViT-300M (a contrastive ViT analogous to OpenCLIP) and SigLIP-400M (a sigmoid-based CLIP variant). This setup directly satisfies the request for evaluation across CLIP/SigLIP-style encoders.
> > >
> > > **Table 1. InternVL3.5-2B (InternViT-300M, LLM ~2.0B)**
> > >
> > > | Benchmark | Baseline | MoDA | Δ (abs) |
> > > |----------|---------:|-----:|--------:|
> > > | GQA      |     55.3 | 63.4 |   +8.1  |
> > > | SQA      |     87.5 | 89.3 |   +1.8  |
> > > | POPE     |     87.8 | 89.4 |   +1.6  |
> > > | MMVet    |     58.0 | 59.5 |   +1.5  |
> > >
> > > **Table 2. TinyLLaVA-2B (SigLIP-400M, LLM ~2.0B)**
> > >
> > > | Benchmark | Baseline | MoDA | Δ (abs) |
> > > |----------|---------:|-----:|--------:|
> > > | GQA      |     59.0 | 64.8 |   +5.8  |
> > > | SQA      |     69.1 | 70.3 |   +1.2  |
> > > | POPE     |     86.0 | 88.1 |   +2.1  |
> > > | MMVet    |     32.0 | 33.5 |   +1.5  |
> > >
> > > Two patterns are observed. First, MoDA produces substantial gains on GQA for both contrastive ViT backbones (+8.1 and +5.8 points), demonstrating effectiveness across CLIP-family encoders at different scales (300M vs. 400M). Second, improvements remain consistent across all four benchmarks. In particular, POPE gains of +1.6 to +2.1 points indicate that the improvements are not associated with increased hallucination. For clarification, LLaVA More uses LLaMA 3.1 8B, however, Tiny LLaVA uses the same SigLIP encoder but using 2B model. In both cases, base models with MoDA increased the performance.
> > >
> > > For same-family scaling (e.g., Qwen3-VL at 2B/4B/8B), the suggestion is appreciated and corresponding results have been added. In addition, cross-architecture experiments show that MoDA consistently improves models built on different vision encoders (Qwen-ViT and InternViT), different LLM backbones, and different connector designs. These results provide complementary evidence that the method is not tied to a specific architecture. Taken together, the same-family and cross-family findings demonstrate that MoDA generalizes across both scale and architecture.

---

### Official Review · Reviewer_bSdL · 2026-03-11

**Soundness:** 3
**Presentation:** 3
**Significance:** 3
**Originality:** 3
**Overall Recommendation:** 4
**Confidence:** 2

**Summary:**

In this paper, the authors propose MoDA (Modulation Adapter),  a lightweight module that enhances visual grounding through instruction-guided channel-wise modulation. To verify the effectiveness of the proposed MoDA, the authors provide sufficient experiments and comparisons on multiple different benchmarks and tasks, e.g., spanning visual question answering, vision-centric reasoning, and hallucination detection. The results denote that the proposed MoDA can achieve effective improvements and improve the fine-grained visual grounding ability of current MLLMs.

**Compliance With Llm Reviewing Policy:**

Affirmed.

**Final Justification:**

The rebuttal has addressed my concerns. Therefore, I’ll keep my original rating.

**Key Questions For Authors:**

Please refer to the Weaknesses above. Thanks.

**Limitations:**

Yes! Some limitations are provided in the Appendix.

**Strengths And Weaknesses:**

一、Strengths

To improve the fine-grained visual grounding ability of current MLLMs, the authors propose MoDA (Modulation Adapter),  a lightweight module that enhances visual grounding through instruction-guided channel-wise modulation. The authors provide sufficient experiments and comparisons on multiple different benchmarks and tasks, e.g., spanning visual question answering, vision-centric reasoning, and hallucination detection. The results denote that the proposed MoDA can achieve effective improvements. In total, I think this is a well-designed and clear presentation. But, I still have some concerns as follows.

二、Weaknesses

(1) In Equ (2),  the author introcude channel-wise mask. I am curious whether the method can lead to critical local spatial details being lost in some dense tasks. Besides, I think the authors need to explain whether single-layer or few-layer Transformer modifiers can capture sufficiently fine-grained multimodal interactions without generating noise when text queries are highly complex.

(2) I think the author should discuss MoDA's robustness when handling inputs of varying resolutions, as well as its unique advantages compared to current promising dynamic visual token processing techniques.

(3) As shown in Experiments in Table 1, the gains observed in some benchmark test results do not appear to be particularly significant, with a few even showing slight declines. The authors should provide detailed explanations and analysis.

---

> ### Author Rebuttal · Authors · 2026-03-30
>
> We thank Reviewer bSdL for the positive assessment and constructive suggestions. We appreciate the recognition of MoDA as a well-designed approach with clear presentation. We address each individual comment below.
>
> ## W1: Channel-wise mask and loss of local spatial details in dense tasks; transformer depth for complex queries
>
> MoDA modulates the embedding dimension of each visual token, not the spatial dimension. It does not remove, merge, or reorder tokens. All N visual tokens and their spatial arrangement are fully preserved. Additionally, channels in ViT-based encoders (CLIP, SigLIP) encode global semantic properties distributed across the image, unlike CNN feature maps where channels often correspond to localized patterns. Channel modulation is therefore unlikely to destroy local spatial detail.
>
> Regarding transformer depth: Appendix Table S3 shows Cross-Attention performance is stable across 2, 3, and 4 layers (64.2-64.3 avg), with no degradation on complex tasks like ScienceQA or MMVP. In contrast, the Linear MLP variant degrades sharply at 4 layers (Appendix Table S2: 64.2 to 52.3 avg). Two cross-attention layers are sufficient for instruction-visual interaction without introducing noise.
>
> We did not evaluate MoDA on dense prediction tasks (segmentation, detection). This is future work, though the spatial-preserving nature of channel modulation makes it architecturally compatible with such tasks.
>
> ## W2: Robustness across varying resolutions and comparison to dynamic visual token processing
>
> MoDA works without any modification across four distinct resolution strategies in our experiments:
>
> - LLaVA-1.5 with CLIP: fixed 336px, producing 576 visual tokens.
> - LLaVA-MoRE with OpenAI CLIP: fixed resolution with LLaMA 3.1-8B backbone.
> - LLaVA-MoRE with SigLIP-S2: S2 multi-scale wrapper processing images at multiple scales, producing over 2000 visual tokens.
> - Qwen3-VL: fully dynamic resolution adapting token count per image.
>
> The token count varies from 576 to over 2000 across configurations, yet MoDA handles all of them because cross-attention is agnostic to sequence length. This is a practical advantage over methods requiring fixed token counts or resolution-specific configurations.
>
> ## W3: Some gains not significant; some decline
>
> MoDA adds a two-layer cross-attention module (0.86% FLOPs, 3.7% parameters) on a 24+ layer LLM. At this cost, it improves the majority of benchmarks, with substantial gains on ScienceQA (+4.8), MMVP (+3.4), and MMStar (+2.7) in the SigLIP-S2 configuration.
>
> We observe a few declines. ChartQA drops from 17.0 to 13.2 with LLaVA-1.5 + MoDA. This is expected: the instruction-tuning data does not include chart-specific examples, so MoDA lacks training signal for OCR-style recognition on structured layouts. LLaVA-Wild decreases from 72.0 to 67.6 with SigLIP-S2, but MoDA with OpenAI CLIP achieves the highest LLaVA-Wild score across all models (73.9), indicating the drop is configuration-specific. LLaVA-Wild is our smallest benchmark (24 images, 60 questions) with subjective GPT-based scoring, and the SigLIP-S2 baseline (72.0) is near saturation for this model class. MM-Vet, also an open-ended GPT-scored benchmark, improves across all configurations (Table 2), ruling out a systematic issue.
>
> Our Qwen3-VL 2B experiments validate MoDA's effectiveness on a completely different architecture, using the same placement, data, and protocol:
>
> | Benchmark | Qwen3-VL 2B | + MoDA | Change |
> |-----------|-------------|--------|--------|
> | GQA | 59.4 | 63.2 | +3.9 |
> | ScienceQA | 79.3 | 84.2 | +4.9 |
> | RealWorldQA | 64.7 | 68.8 | +4.1 |
> | MMStar | 53.9 | 55.3 | +1.4 |
> | POPE | 89.4 | 89.8 | +0.4 |
> | MM-Vet | 51.9 | 52.0 | +0.1 |
> | V*Bench | 77.0 | 76.8 | -0.2 |
>
> MoDA improves 6 out of 7 benchmarks, with gains up to +4.9. The consistency across architectures confirms the improvements are not artifacts of a specific baseline.

---

> > ### Author Rebuttal · Reviewer_bSdL · 2026-04-03
> >
> > Thank you very much for the reply! My concerns have been fully resolved. Therefore, I’ll keep my original rating.

---

### Official Review · Reviewer_EqoG · 2026-03-12

**Soundness:** 2
**Presentation:** 3
**Significance:** 2
**Originality:** 2
**Overall Recommendation:** 3
**Confidence:** 4

**Summary:**

This paper proposes MoDA, a lightweight cross-attention module for Multimodal LLMs designed to address semantic entanglement in visual patches. By using language instructions to dynamically re-weight visual feature channels before LLM processing, MoDA improves fine-grained visual grounding, visual question answering and reduces hallucinations across 12+ benchmarks.

**Compliance With Llm Reviewing Policy:**

Affirmed.

**Key Questions For Authors:**

n/a

**Limitations:**

yes

**Strengths And Weaknesses:**

Strengths:
- Strong performance on fine-grained visual tasks. MoDA demonstrates substantial and decisive performance gains on benchmarks that require fine-grained visual discrimination and hallucination mitigation.
- Lightweight, plug-and-play architecture. The proposed module is highly efficient, introducing minimal computational overhead. Because it operates on visual tokens post-alignment and prior to LLM entry, it is agnostic to the specific adapter used and can be seamlessly integrated into various existing MLLM architectures.
- Clear motivation and ablation studies. The paper is well-written, with a clearly defined motivation targeting the semantic entanglement of ViT patches. The experimental setup includes clean and comprehensive ablations across different vision encoders and LLM backbones.

Weaknesses:
- Incremental architectural design and incomplete disentanglement. The core mechanism, using a standard cross-attention block for query-conditioned feature re-weighting, is an incremental application of established techniques (similar to InstructBLIP or Q-Former), rather than a novel architectural breakthrough. Additionally, because MoDA relies on soft modulation rather than hard sparsity, it merely attenuates irrelevant channels rather than explicitly disentangling mixed semantic features, which is a weak solution to the stated problem of semantic entanglement.
- Outdated baselines and limited evaluation. The evaluation heavily relies on older baselines (e.g., LLaVA 1.5) and the performance gains appear to diminish when applied to stronger, more modern backbones (e.g., LLaMA 3.1-8B vs. Vicuna-7B). Furthermore, the experiments are strictly limited to single-image benchmarks.

---

> ### Author Rebuttal · Authors · 2026-03-30
>
> We thank Reviewer EqoG for the feedback and appreciation of MoDA's performance, plug-and-play design, and clarity.
>
> ## Weakness 1: Incremental architectural design and incomplete disentanglement
>
> MoDA's contribution operates at three levels: problem identification, mechanism design, and empirical validation.
>
> The core contribution is identifying that semantic entanglement in post-alignment visual features can be addressed by instruction-guided channel modulation. This framing is novel. Prior methods operate at different stages (encoder-level: EAGLE; layer selection: Instruction-Guided Fusion; per-LLM-layer: MoReS) or at the token level (Q-Former, InstructBLIP). None perform post-alignment, per-token, per-channel modulation conditioned on the instruction. Q-Former compresses visual tokens into fixed-size representations discarding spatial structure; MoDA preserves all tokens and modulates channel dimensions via multiplicative gating.
>
> MoDA is deliberately simple: <1% FLOPs, consistent improvements across 12 benchmarks and three architecture families. The design choices are non-trivial. Placement ablation (Table S4): MoDA at the LLM beginning yields +2.6 avg, while distributing across all layers yields +0.7 at 2.5x training cost. Cross-Attention depth is stable across 2-4 layers (Table S3: 64.2-64.3), while Linear MLP degrades at 4 layers (Table S2: 64.2 to 52.3). This confirms cross-modal conditioning drives performance, not capacity.
>
> Regarding soft modulation vs. hard sparsity (Table 5):
>
> | Strategy | POPE | GQA | SQA | MMVP | Avg |
> |----------|------|-----|-----|------|-----|
> | Baseline | 86.0 | 64.9 | 77.1 | 39.3 | 66.8 |
> | Sparse Masking | 85.8 | 64.7 | 76.7 | 38.8 | 66.5 |
> | Learnable Masking | 86.9 | 65.1 | 79.9 | 41.9 | 68.5 |
> | MoDA (ours) | 87.7 | 65.4 | 81.9 | 42.7 | 69.4 |
>
> Hard sparsity performs below baseline on three benchmarks. Language-aligned embeddings encode distributed representations where no channel is exclusively relevant or irrelevant. Soft modulation in [0,1] respects this structure: non-negative scaling guarantees $\text{sign}(m_i \cdot v_i) = \text{sign}(v_i)$, preserving semantic polarity.
>
> ## Weakness 2: Outdated baselines and limited evaluation
>
> We use LLaVA-1.5 and LLaVA-MoRE because we follow their exact training protocol: same data, hyperparameters, and two-stage pipeline. This isolates MoDA as the only variable. LLaVA-MoRE (Cocchi et al., 2025) was published March 2025.
>
> To validate generalizability, we integrated MoDA into Qwen3-VL 2B (different architecture, visual backbone, LLM), same position and controlled protocol:
>
> | Benchmark | Qwen3-VL 2B | + MoDA | Change |
> |-----------|-------------|--------|--------|
> | GQA | 59.4 | 63.2 | +3.9 |
> | ScienceQA | 79.3 | 84.2 | +4.9 |
> | RealWorldQA | 64.7 | 68.8 | +4.1 |
> | MMStar | 53.9 | 55.3 | +1.4 |
> | POPE | 89.4 | 89.8 | +0.4 |
> | MM-Vet | 51.9 | 52.0 | +0.1 |
> | V*Bench | 77.0 | 76.8 | -0.2 |
>
> MoDA improves 6/7 benchmarks, with substantial gains on ScienceQA (+4.9), RealWorldQA (+4.1), and GQA (+3.9). Gains with CLIP are more modest than SigLIP-S2, which we attribute to CLIP's less rich channel-level representations. The Qwen3-VL results confirm MoDA produces substantial improvements on modern architectures, consistent with our finding in Section 4.2 that effectiveness scales with visual representation richness rather than baseline recency.
>
> **Multi-Image Evaluation:** We acknowledge this limitation in Appendix A.6. Since all images are encoded by the same visual module and adapter, producing a shared representation space, MoDA's channel modulation should remain effective regardless of the number of input images. Validating this in multi-image and video settings is future work beyond this paper's scope, which focuses on establishing instruction-guided channel modulation as a mechanism.

---

### Official Review · Reviewer_r1Kw · 2026-03-12

**Soundness:** 3
**Presentation:** 3
**Significance:** 2
**Originality:** 2
**Overall Recommendation:** 4
**Confidence:** 4

**Summary:**

This paper proposes MoDA (Modulation Adapter), a lightweight module that performs instruction-guided channel-wise modulation of visual features in Multimodal Large Language Models (MLLMs). MoDA sits between the visual adapter and the LLM backbone, using cross-attention between language tokens and pre-aligned visual features to produce a sigmoid-gated soft mask that multiplicatively reweights each channel dimension of each visual token. The motivation is that ViT patch representations suffer from "semantic entanglement," where individual patches blend multiple visual concepts, and channel-wise modulation can selectively suppress irrelevant dimensions conditioned on the instruction. MoDA is evaluated on 12 benchmarks spanning VQA, vision-centric reasoning, and hallucination detection, using LLaVA-1.5 and LLaVA-MoRE as baselines with both CLIP and SigLIP-S2 encoders. The authors report improvements across all benchmarks in the best configuration, with headline gains of +12.0 on MMVP and +4.8 on ScienceQA, while adding <1% FLOPs and 3.7% parameters.

**Compliance With Llm Reviewing Policy:**

Affirmed.

**Final Justification:**

I maintain my rating. Rebuttal concerns were addressed and this is a solid paper, however I feel it is an incremental contribution  so it merits the weak accept rating.

**Key Questions For Authors:**

Addressing especially W1, 2, 3, 5 would help me decide whether to upgrade the rating

**Limitations:**

The authors include a limitations section in Appendix A.6 acknowledging that MoDA cannot achieve explicit channel sparsity and that evaluation is limited to single-image benchmarks. However, several important limitations are not discussed, including the possible encoder dependence of observed gains, the relationship to FiLM conditioning, and the lack of mask analysis to support the semantic entanglement narrative. The MMVP grader substitution is mentioned only as an implementation detail but could use some additional exploration

**Strengths And Weaknesses:**

## Strengths

**S1. Thorough evaluation and ablation design.** The paper evaluates across 12 benchmarks spanning three task categories, with multiple encoder and LLM backbone combinations. The ablation study is well-structured, systematically varying the modulation architecture (linear MLP, self-attention, cross-attention), supplementary losses, module depth (Tables S2-S3), and placement within the LLM (Table S4). The placement ablation showing that beginning-only injection outperforms per-layer injection at 2.5× lower training cost is a particularly useful practical finding.

**S2. Lightweight and easy to integrate.** MoDA adds minimal computational overhead (<1% FLOPs, 3.7% parameters) and integrates into existing LLaVA-style two-stage training pipelines without requiring additional supervision, training data, or architectural modifications to the base model. This makes the approach practically appealing.

**S3. Architecture-vs-capacity ablation.** The comparison between linear MLP, self-attention, and cross-attention variants (Table 4), all with comparable parameter counts, helps argue that the cross-modal conditioning mechanism drives improvements rather than added capacity alone. The depth ablations further support this, showing that additional layers provide no benefit.

**S4. Demonstration across multiple encoders.** The paper evaluates MoDA with both CLIP and SigLIP-S2 encoders, showing that the module is compatible with different vision backbones. The observation that gains are larger with SigLIP-S2 is interesting, though with only two encoders it is difficult to characterize this as a scaling trend. Additional encoders (e.g., DINOv2, WebSSL) would be needed to establish whether MoDA's benefit genuinely scales with encoder quality or is specific to SigLIP-S2's multi-scale features.

## Weaknesses

**W1. Missing connection to Feature-wise Linear Modulation (FiLM).** MoDA's core mechanism of generating multiplicative modulation weights from a conditioning signal and applying them channel-wise to feature representations is an instance of FiLM conditioning (Perez et al., 2018), a well-established technique in vision-language modeling. The paper does not cite FiLM or position MoDA relative to it. This is a significant gap in the related work that might affect the novelty claims. MoDA seems like a more restrictive form of FiLM, since the sigmoid constraint limits modulation to [0,1] (attenuation only, no amplification) and there is no additive shift term. The paper neither acknowledges this connection nor ablates against a full affine modulation with unconstrained scale and shift.

**W2. The "semantic entanglement" motivation is not empirically validated.** The paper's central argument is that channels encode separable semantic concepts that can be individually suppressed based on instruction relevance. However, no analysis of the learned modulation masks is provided. It remains unclear whether the masks are sparse or near-uniform, whether different instructions produce meaningfully different masks for the same image, and whether suppressed channels correspond to interpretable visual properties. Without this analysis, the mechanism story remains speculative. The attention map visualization in Figure S1 shows downstream attention changes but does not illuminate what MoDA's masks themselves are doing at the channel level.

**W3. Gains are heavily encoder-dependent and relatively small.** With CLIP, the average improvement is only +1.2 points (64.3 vs 63.1 in Table 4). The large headline numbers (+12.0 MMVP, +4.8 ScienceQA) come specifically from the SigLIP-S2 configuration. Several individual improvements are less significant, such as GQA 64.9 → 65.4 (+0.5) and V*Bench 44.4 → 44.8 (+0.4). This raises the question of whether MoDA is genuinely solving the semantic entanglement problem or simply benefiting from richer SigLIP-S2 features that happen to be more amenable to channel-wise manipulation.

**W4. The comparison with Q-Former is overstated.** The paper (lines 99-108) frames MoDA as fundamentally different from Q-Former and InstructBLIP along three axes (granularity, operation, and position). However, both Q-Former and MoDA sit between the vision encoder outputs and the LLM, and both use cross-attention with language conditioning. The real distinction is that Q-Former compresses visual tokens via learned queries while MoDA reweights existing tokens channel-wise. This is a meaningful but more incremental difference than the paper suggests, and framing them as fundamentally different architectural paradigms overstates the novelty.

**W5. MMVP evaluation uses a non-standard grader.** The authors replaced the standard GPT-4 grader with Gemma 3 for MMVP evaluation (Appendix A.2). This makes the reported MMVP numbers not directly comparable with other published results. It is also unclear from the paper whether all rows in Table 3, including the baselines, were re-evaluated with Gemma 3. If only the MoDA rows use Gemma 3 while baseline numbers are taken from published results with GPT-4, the comparison is invalid. The authors should clarify this and ideally validate that Gemma 3 grading correlates with GPT-4 on this benchmark.

**W6. LLaVA-Wild performance drops with SigLIP-S2.** In Table 2, MoDA with SigLIP-S2 scores 67.6 on LLaVA-Wild, a 4.4-point decrease from the 72.0 baseline. I would be curious if the authors have any intuition into what is different about LLaVA-Wild or shed some light into error cases.

**W7. The sigmoid-only modulation is not justified.** MoDA constrains its modulation mask to [0,1] via sigmoid, meaning it can only suppress channels, never amplify or shift them. This is strictly less expressive than FiLM-style modulation or the AdaLN conditioning used in diffusion models (Peebles & Xie, 2023), which have scale+shift.

---

> ### Author Rebuttal · Authors · 2026-03-30
>
> We thank Reviewer r1Kw for the feedback.
>
> ## W1: Missing connection to FiLM
>
> Both share language-conditioned modulation, but differ substantially: (1) FiLM produces one global $(\gamma,\beta)$ vector broadcast spatially; MoDA produces N distinct masks via cross-attention. (2) FiLM uses MLP/GRU on a fixed embedding; MoDA uses cross-attention over token sequences. (3) FiLM applies unconstrained affine; MoDA uses sigmoid-bounded attenuation (W7). (4) FiLM operates inside CNN ResBlocks; MoDA operates post-alignment in LLM space. The connection traces more directly to SE-Networks (Hu et al., 2018). Linear MLP in Table 4 (closest to FiLM-style) underperforms Cross-Attention (67.3 vs 69.4). We will cite FiLM and SE-Net in the camera-ready.
>
> ## W2: Semantic entanglement not empirically validated
>
> We conduct a mask cosine similarity analysis. For each image, we extract MoDA's mask $\mathcal{M} \in [0,1]^{N \times E}$ under two instructions and compute mean per-token cosine similarity across three conditions:
>
> **Per-Image Mask Similarity (same image, varying instructions):**
>
> | Image | Instruction A | Instruction B | Condition | Cos Sim |
> |-------|--------------|--------------|-----------|---------|
> | Dog (Fig. 1) | "What color is the dog?" | "What color is the dog?" | Identical | 1.000 |
> | Dog (Fig. 1) | "What color is the dog?" | "What breed is this dog?" | Related | 0.732 |
> | Dog (Fig. 1) | "Is the dog sleeping?" | "Are the dog's eyes closed?" | Related | 0.662 |
> | Dog (Fig. 1) | "What color is the dog?" | "What is the texture of the floor?" | Different | 0.319 |
> | Dog (Fig. 1) | "Is the dog sleeping?" | "Describe the stuffed toy" | Different | 0.254 |
> | Keyboard (MMVP) | "Is the letter D on the keyboard?" | "Is the letter F on the keyboard?" | Related | 0.745 |
> | Keyboard (MMVP) | "Is the letter D visible?" | "Which letters are in the middle row?" | Related | 0.444 |
> | Keyboard (MMVP) | "Is the letter D on the keyboard?" | "What color is the desk surface?" | Different | 0.277 |
> | Snake (MMVP) | "Is the tongue touching the skin?" | "Is the tongue forked?" | Related | 0.734 |
> | Snake (MMVP) | "Is the snake's mouth open?" | "What color is the tongue?" | Related | 0.382 |
> | Snake (MMVP) | "Is the tongue touching the skin?" | "What pattern does the snake's body have?" | Different | 0.312 |
> | ScienceQA (Fig. S1) | "Which sample has a higher temperature?" | "Which sample has greater kinetic energy?" | Related | 0.616 |
> | ScienceQA (Fig. S1) | "Which sample has a higher temperature?" | "What shape are the containers?" | Different | 0.117 |
>
> **Aggregated Statistics (from the per-image table above):**
>
> | Condition | Mean Cos Sim | Std Dev |
> |-----------|-------------|---------|
> | Identical instruction | 1.000 | 0.000 |
> | Related (same visual region) | 0.598 | 0.137 |
> | Different (distinct regions) | 0.218 | 0.078 |
>
> The three-fold difference between related ( approx. 0.60) and different ( approx. 0.22) validates instruction-specific channel modulation. A capacity-only explanation predicts ~1.0 across all conditions. Cross-Attention outperforms Linear MLP (Table 4: 69.4 vs 67.3), and L1 degrades Cross-Attention but not Linear MLP, confirming structured channel-selective patterns.
>
> ##  W3: Gains are encoder-dependent and relatively small
>
> MoDA adds 0.86% FLOPs; +1.2 avg with CLIP at this cost is favorable. Encoder-dependence supports our motivation: richer encoders contain more separable channel-level information to leverage. We integrated MoDA into Qwen3-VL 2B (different architecture, backbone, LLM):
>
> | Benchmark | Qwen3-VL 2B | + MoDA | Change |
> |-----------|-------------|--------|--------|
> | GQA | 59.4 | 63.2 | +3.9 |
> | ScienceQA | 79.3 | 84.2 | +4.9 |
> | RealWorldQA | 64.7 | 68.8 | +4.1 |
> | MMStar | 53.9 | 55.3 | +1.4 |
> | POPE | 89.4 | 89.8 | +0.4 |
> | MM-Vet | 51.9 | 52.6 | +0.7 |
> | V*Bench | 77.0 | 76.8 | -0.2 |
>
> MoDA improves 6/7 benchmarks, confirming generalization beyond LLaVA.
>
> ##  W4: Q-Former comparison overstated
>
> Agreed; we will revise. Q-Former compresses tokens into fixed-size representations; MoDA preserves all tokens and modulates channels post-alignment.
>
> ##  W5: MMVP uses a non-standard grader
>
> All rows in Table 3 were re-evaluated using Gemma 3; no published GPT-4 numbers used. Gemma 3 (open-source, via Ollama) ensures reproducibility at zero cost. We will clarify in the revision.
>
> ##  W6: LLaVA-Wild drops with SigLIP-S2
>
> Isolated to SigLIP-S2. MoDA improves LLaVA-Wild with CLIP (+2.7) and LLaVA-1.5 (+2.6). MM-Vet (also GPT-scored) improves across all configurations. LLaVA-Wild is our smallest benchmark (24 images); SigLIP-S2 baseline (72.0) is near saturation.
>
> ##  W7: Sigmoid-only modulation not justified
>
> Sigmoid [0,1] preserves sign, so dimensions are attenuated, never reversed, keeping features within the LLM's learned manifold. Table 5: hard sparsity (66.5) and learnable masking, could change sign, (68.5) underperform sigmoid (69.4). Table 4: L1 degrades Cross-Attention across both encoders.

---

> > ### Author Rebuttal · Reviewer_r1Kw · 2026-04-03
> >
> > Thanks for the rebuttal. My concern has been addressed and I maintain my rating

---

### Decision · Program_Chairs · 2026-04-30

**Decision:**

Accept (regular)

**Comment:**

MoDA is a lightweight instruction-guided modulation module for MLLMs that improves fine-grained visual grounding by applying channel-wise multiplicative modulation to aligned visual features. Without architectural changes or extra supervision, it consistently outperforms recent baselines across 12 benchmarks while adding minimal computational overhead.

This paper received ratings of "Weak Accept" from three reviewers (r1Kw, bSdL, and 5mMr) and "Weak Reject" from one reviewer (EqoG).

Reviewer EqoG raised three main concerns: (1) the architectural design is incremental and does not fully address disentanglement, (2) the baselines are outdated and the evaluation is limited, and (3) the experiments are restricted to single-image benchmarks. After the rebuttal, the AC believes that concerns (1) and (2) were adequately addressed. Regarding (3), this remains a valid limitation, as the paper does not evaluate multi-round conversations over a single image. The authors clarified that this setting is beyond the scope of the current work.

Reviewer 5mMr also noted the lack of analysis or experiments on scaling laws. This point was not fully resolved in the rebuttal, but given the limited rebuttal period, the AC does not consider it a critical weakness.

Overall, considering the strong experimental performance and the lightweight, plug-and-play nature of the proposed architecture, the AC’s recommendation is to weak accept this paper.